# Assessing Climate Change Impact on Cropland Suitability in Kyrgyzstan: Where Are Potential High-Quality Cropland and the Way to the Future

Sugyeong Park [1], Chul-Hee Lim [2], Sea Jin Kim [3], Erkin Isaev [4], Sol-E Choi [5], Sung-Dae Lee [6] and Woo-Kyun Lee [5,*]

1  OJeong Resilience Institute (OJERI), Korea University, 145 Anamro, Seongbukgu, Seoul 02841, Korea; synergyeong@gmail.com
2  College of General Education, Kookmin University, 77 Jeongneungro, Seongbukgu, Seoul 02707, Korea; clim@kookmin.ac.kr
3  Risk Advisory, Deloitte Anjin LLC, One IFC, 10, Gukjegeumyung-ro, Yeongdeungpo-gu, Seoul 07326, Korea; bluegulcy@gmail.com
4  Mountain Societies Research Institute, University of Central Asia, 138 Toktogul Street, Bishkek 720001, Kyrgyzstan; erkin.isaev@ucentralasia.org
5  Department of Environmental Science and Ecological Engineering, Korea University, 145 Anamro, Seongbukgu, Seoul 02841, Korea; pine0630@gmail.com
6  Yumkwang High School, 9, Wolgyero 45ga-gil, Nowongu, Seoul 01874, Korea; illskys@kakao.com
*  Correspondence: leewk@korea.ac.kr

**Abstract:** Climate change is one of the greatest challenges in Kyrgyzstan. There have been negative spillover effects in agriculture. This study aims to assess the climate change impacts on cropland suitability in Kyrgyzstan. We used the random forest algorithm to develop a model that captures the effects of multiple climate and environment factors at a spatial resolution of 1 km$^2$. The model was then applied in the scenario analysis for an understanding of how climate change affects cropland distribution. The potential high-quality cropland was found to be included in existing croplands, while the remaining were distributed around the Chu-Talas valley, the Issyk-kul area, and the Fergana valley. These potential high-quality croplands comprise grasslands (47.1%) and croplands (43.7%). In the future, the potential high-quality cropland exhibited inland trends at the periphery of original cropland category, with grassland and cropland as the primary land components. Due to climate change, potential high-quality cropland is expected to gradually reduce from the 2050s to the 2070s, exhibiting the largest reduction in potential high-quality areas for the Representative Concentration Pathway 8.5 scenario. Therefore, the short- and long-term adaptation strategies are needed for prioritizing the croplands to ensure food security and agricultural resilience.

**Keywords:** cropland suitability; random forest model; scenario analysis; climate change adaptation; Kyrgyzstan

## 1. Introduction

Climate change is one of the greatest challenges in Kyrgyzstan, which is reported to be the third most vulnerable country in Eastern Europe and Central Asia, mainly owing to the climate-sensitive agricultural system and lack of adaptive capacity [1]. Kyrgyzstan is threatened with glacier melting and a lack of freshwater balance, which are accelerated by global warming. Moreover, the country has been suffering from aridity and drought in its mountain pastures in recent years, which are triggered by frequent extreme heat and abnormal rainfall events [2]. According to the simulated climate scenarios in a neighboring country, maize yield is projected to be decrease in current maize-producing areas under climate change, but future yield can be increased with the irrigation and planting adaptation strategies [3].

To combat climate change, Kyrgyzstan has submitted National Communications to the United Nations Framework Convention on Climate Change (UNFCCC) in 2017, presenting therein the importance of agricultural roles in the country and the uncompromising necessity of a proactive approach to agricultural resilience. According to reports, agriculture has improved the economics of the country by producing 1/5 of its gross domestic product (GDP). However, unstable crop yields have caused decreases in food provision, causing it to fluctuate each year. The periods of late spring, early autumn, and high temperature preclude stable cultivation [4]. It has been reported that the economic damage in terms of water resource and agriculture, determined by estimating the amount of economic losses in the absence of appropriate adaptation efforts, has reached approximately 788 million USD, accounting for 64% of total economic damage to the country. Thus, it is necessary for Kyrgyzstan to evaluate cultivation suitability and suggest climate adaptation strategies focusing on land sustainability and ecological impacts in the agricultural sector [5].

Agricultural adaptation is proposed in the national policy document, "Priority Directions for Climate Change Adaptation (PDCCA)". In this document, the government approved priority measurements in the agricultural sector, such as the improvement of agricultural infrastructure for better adaptation to the negative impacts of climate change. The improvement of agricultural infrastructure implies the rehabilitation of the existing and construction of new water management facilities and planting of forest plantations to combat coastal erosion [6]. For example, the recently published draft of the State Irrigation Development Program (2017–2026) is aimed at solving the problem of the efficient use of water resources. The program aims to provide new irrigated land for rural residents to grow agricultural products, improve the socio-economic situation of the regions, and solve issues of food security. The program will allow for the allocation of about 850 million USD to introduce 65,500 hectares of irrigated land, increase the water supply on 51,000 hectares, 9500 hectares to be transferred from pumping irrigation to gravity irrigation, and 50,000 hectares to improve ameliorative condition [7].

Although improvements to agricultural infrastructure are considered as the one of the most common approaches for cropland management, many agricultural households in Kyrgyzstan do not frequently employ sustainable land management technologies, compared that 60% of the agricultural households in neighboring countries use sustainable land management technologies such as integrated soil fertility management, drip irrigation, and the use of portable chutes in sloping areas [8]. There is no doubt that sustainable land management like the integrated soil fertility management and irrigation techniques should be applied in Kyrgyzstan.

In terms of land use condition, crops are cultivated mostly in the valley and foothill regions. Pastures account for more than half the area of the land, followed by the arable lands and hayfields. This land use modality was formed by a consequence of transition time from the Soviet Union. The ways of land use have been changed mainly from sheep production, into crop production during transition time. Kyrgyzstan in crop production has made an enormous accomplishment, expanding wheat area from 250 to 550 thousand hectares by 1997 and increased in share of dry beans production in the market [9,10]. Contrarily, the livestock industry has long remained a neglected sector, and faced the trouble to improve rangeland management, so it is reported that the integrated crop and livestock production need to be established [11].

Considering that agriculture is a climate-sensitive sector, and varies with adaptive policies and land use strategies, it is significant to determine priority areas for applying adaptive policies and effective strategies. To identify the priority areas, we used random forest model (RF), which is known as an ensemble-learning algorithm for classification and regression. In addition, it generates many individual decision trees on randomly selected bootstrap samples with low bias and variance [12,13]. The RF algorithm has been used extensively, such as in remote sensing, ecology, and climate change studies [14,15]. A previous study proposed the habitat suitability of Pinus sylvestris using several machine learning techniques [16]. Moreover, the maximum entropy and RF models have been

applied in a previous study to predict the spatial distribution of the probability of forest fires [17].

Hence, this study aims to identify potential high-quality croplands based on the present high-quality cropland and suggest potential high-quality croplands for the future by using the Representative Concentration Pathway (RCP) 4.5 and 8.5 scenarios for the 2050s and 2070s, respectively, and then prioritize target areas for applying short- and long-term adaptation strategies. The study used the random forest model of machine learning that enables us to analyze potential high-quality croplands and spatial distributions in the future by using different time scales [18].

## 2. Materials and Methods

### 2.1. Research Area

The research area is Kyrgyzstan, located in Central Asia between latitudes of 39 and 44° N and longitudes of 69 and 81° E (Figure 1). The country comprises an area of approximately 200,048 km². The climate of Kyrgyzstan is regionally heterogeneous. The landscapes of Kyrgyzstan can be grouped into four climate zones [19]. The valley sub-mountain zone is characterized by a hot summer, snowless and temperate winters, and almost zero precipitation. The mountain zone is temperate with warm summers as well as cold and snowy winters. The high-mountain zone is cooler in the summer and has relatively cold and snowless winters, with temperatures ranging from below 0 to 16 °C. The nival belt zone has a polar climate and is covered with snowfields and glaciers [20]. According to the Köppen climate classification, cold in winter, dry and hot in summer (Dsa), and cold arid, steppe, and cold climate (Bsk) are the most prevalent characteristics of these aforementioned climate zones [21].

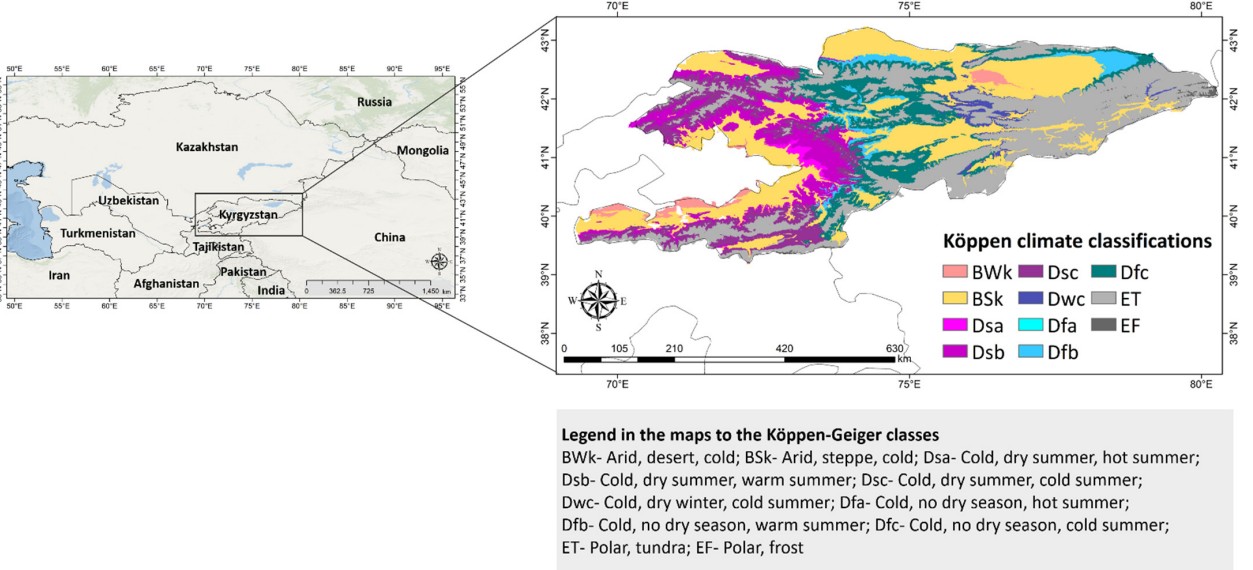

**Figure 1.** Map of the study area with the locations of Central Asia and Köppen climate classification of Kyrgyzstan.

Geographically, the highest peak is 7439 m in the Kakshaal-Too range, where the shared border with China is located, consisting of glacial and sub-glacial areas. On the contrary, the lowest point is 132 m above sea level, which is located along the Kara Darya River in the Fergana Valley [22]. There is a high percentage of grassland, accounting for approximately 56% (112,013 km²) of the total land area, and bareland, which covers 13.1% (26,268 km²); moreover, the croplands, snowy terrain, and forests comprise 8.1% (16,188 km²), 9.9% (191,843 km²), and 4.6% (9272 km²), respectively, and the other lands account for 8.3% (16,482 km²) of the total land area, as estimated by the 2010 Global Land Cover data at a 30-m resolution. For this study, Kyrgyzstan was divided into a grid with a spatial resolution of 1 km².

*2.2. Data*

2.2.1. Time-Series NDVI Data

The satellite-derived vegetation indices have been widely used for crop monitoring and yield estimate in previous research [23,24]. To be specific, the normalized difference vegetation index (NDVI) has been proven to be strong correlations with wheat, paddy rice, and corn [25]. The main crops in Kyrgyzstan are spring wheat, winter wheat, and maize. According to the crop calendar, each crop actively sows, grows, and is harvested throughout the year. For winter wheat, it is sown from the end of August to the beginning of October and grew from October through May. Winter wheat harvested from June to the beginning of August [26].

Thus, the NDVI time series data from the National Oceanic and Atmospheric Association satellite data has been used in this study for detecting high-quality cropland [27]. Satellite time series of the MODIS NDVI 16-day global 250-m data covering the period from 2014 to 2018 were requested from the University of Natural Resources and Applied Life Sciences (BOKU) (http://ivfl-info.boku.ac.at (accessed on 7 August 2020)). The monthly NDVI products from 2014 to 2018 were rebuilt based on the maximum value using a cell statistic tool in ArcGIS, since MODIS vegetation indices produced on 16-day intervals, having at least 2 tiles of NDVI image in the month [28,29]. Annual maximum NDVIs (2014 to 2018) were masked by the extent of the Kyrgyzstan boundary, respectively. In this study, we created maximum cropland's NDVI via annual maximum NDVIs (2014–2018) to use as a proxy of agricultural productivity.

2.2.2. Climate Data

We listed climate and environmental variables to understand the impact of climate change on cropland suitability that have high possibilities of determining cropland [30]. The climate variables were set up 19 climate data and 5 indexes that have been proven to have an influence on agriculture (Table 1). These 19 climatic data were obtained from the climatologies at a high resolution for the earth's land surface areas (CHELSA) dataset at a 30-arcsec resolution to ensure the accuracy of climatological data, and these were generated based mainly on the monthly average temperature and precipitation, as gathered from meteorological stations for the 34-year period from 1979 to 2013, and interpolated in relation to the global surface [31]. In addition to climate variables, we generated 5 additional indexes with regard to the extent of data availability based on the above climate data, as used widely in previous studies. To project future cropland suitability, considering the climate in the future, we compared two cases based on the two different climate scenarios, RCP4.5 and RCP8.5. The future climate data was obtained from the CHELSA dataset, which employed the HadGEM2-AO model under the RCP2.6, RCP4.5, RCP6.0, and RCP8.5 scenarios. The simulation periods were 2041 to 2060 (2050s) and 2061 to 2080 (2070s), respectively.

**Table 1.** List of all climate and environmental variables.

| Climate Variables | | Enviornmental Variables |
|---|---|---|
| Bio1, Annual Mean Temperature (°C) | Bio13, Precipitation of Wettest Month (mm) | Slope |
| Bio2, Mean Diurnal Range (°C) | Bio14, Precipitation of Driest Month (mm) | Topographic Wetness Index (TWI) |
| Bio3, Isothermality (°C) | Bio15, Precipitation Seasonality (mm) | Ecological Land Units (ELU) |
| Bio4, Temperature Seasonality (°C) | Bio16, Precipitation of Wettest Quarter (mm) | |
| Bio5, Max Temperature of Warmest Month (°C) | Bio17, Precipitation of Driest Quarter (mm) | |
| Bio6, Min Temperature of Coldest Month (°C) | Bio18, Precipitation of Warmest Quarter (mm) | |
| Bio7, Temperature Annual Range (°C) | Bio19, Precipitation of Coldest Quarter (mm) | |
| Bio8, Mean Temperature of Wettest Quarter (°C) | PEI, Precipitation Effectiveness Index | |

**Table 1.** *Cont.*

| Climate Variables | | Enviornmental Variables |
|---|---|---|
| Bio9, Mean Temperature of Driest Quarter (°C) | Warmth Index (WI) | |
| Bio10, Mean Temperature of Warmest Quarter (°C) | Aridity Index (AI) | |
| Bio11, Mean Temperature of Coldest Quarter (°C) | Climate Moisture Index (CMI) | |
| Bio12, Annual Precipitation (mm) | Evapotranspiration Index (PET) | |

### 2.2.3. Environmental Data

With regard to the environmental variables, 3 variables were used in this study: Slope, Topographic Wetness Index (TWI), and Ecological Land Units (ELUs). Slope and TWI were drawn from the Digital Elevation Model (DEM) accessed from the 30-m SRTM website (http://dwtkns.com/srtm30m/ (accessed on 9 June 2020)), which qualified topographic control on hydrological processes [32]. The ELUs were obtained from global ecosystem data in the United States Geological Survey (https://www.usgs.gov (accessed on 30 June 2020)) developed by Sayrem R in 2014; it presents distinct bioclimates, landforms, lithologies, and land covers, and it has been widely used in several studies for the purposes of classifying ecosystem regions [33,34], determining vegetation patterns [35,36], and establishing a national ecological framework [37]. As edaphic factors, slope, and TWI have been used in previous studies for predicting soil types [38,39]. Furthermore, ELUs were used to restrictively reflect the soil moisture data for the model due to the lack of high-resolution soil moisture data in the target country. In terms of predictions for the future, this study assumed that the environmental variables would default when current climate is replaced by climate scenarios.

### 2.3. Method

### 2.3.1. Classification of High-Quality Croplands

In this study, we classified high-quality croplands to be used as the labeled data for the random forest model. High-quality croplands refer to areas where agricultural productivity is relatively higher than other areas [40,41]. To identify the high-quality croplands from annual maximum NDVIs (2014–2018), we performed the following cell analysis. First the 2010 Global Land Cover data was used to define the extent of land use category in the Kyrgyzstan, which was accessed from their website (http://www.globallandcover.com (accessed on 16 September 2019)). Land cover maps were resampled from a 30-m to 250-m resolution to match with the spatial resolution of NDVI. Second, the extent of cropland were masked from the annual maximum NDVIs, respectively. Thereafter, annual cropland's NDVI was generated from 2014 to 2018. Third, we established the maximum cropland's NDVI by analyzing the annual cropland's NDVI based on the maximum value throughout cell statistics analysis. However, the overlapping of the forest edge near the cropland was unavoidable due to the resolution gaps between annual maximum NDVIs (250 m) and land cover map (30 m). To avoid overestimation, the forest edges were removed from the maximum cropland's NDVI [42]. As for other land classes like grassland, shrubland, wetland, water, Tundra, urban, bareland, and snow, the NDVI value of these classes are lower than cropland. The impacts of these classes were removed while analyzing cell statistic based on the maximum value. Finally, the maximum cropland's NDVI were created without the impact of the forest, which becomes a universal set (*U*) for classifying high-quality croplands. To classify high-quality croplands (*H*) as a labeled data, we set the threshold throughout the universal set (*U*).

$$H = \{x \in U \mid x \text{ is in top 5th percent}\} \tag{1}$$

where *U* is a universal set of the maximum cropland's NDVI. The number of elements is 65,196. The maximum cropland's NDVI was recorded as 0.905, while the minimum value of was 0.111, and the mean value is approximately 0.661. According to Equation

(1), high-quality croplands (*H*) are the top 5 percent of value of the maximum cropland's NDVI (*U*). Considering the maximum value and minimum value, the upper 5 percent of the maximum cropland's NDVI (*U*) was 0.802. The threshold was set based on two criteria: To detect the cropland with high agricultural productivity and to ensure sufficient cell numbers from universal set (*U*) for modeling the training data. Thus, the high-quality croplands (*H*) considered as equal and higher than 0.802. After selecting the values of high-quality croplands, the raster data transformed as point shapefile to indicate the location of high-quality cropland with the coordinate of WGS 1984.

### 2.3.2. Classification of Potential High-Quality Croplands

The potential high-quality croplands refer to the land that has not been converted but it is suitable to be used as high-quality cropland. In this study, we predict the potential high-quality cropland based on high-quality croplands using the random forest model.

$$C = \{x_i | 0 \leq x_i \leq 1, \ x_i \leq x_j, \ i < j, \ i = 1 \dots .n\}. \tag{2}$$

In Equation (2), the results of the random forest model could present cropland suitability (C) that range from 0 to 1, where 1 pertains to a high probability of being a potential high-quality cropland, whereas 0 pertains to not being unsuitable for a cropland. Cropland suitability is arranged by ascending order. Since it is important to classify where the potential high-quality croplands will be considered, this study set the thresholds on the consideration of potential high-quality croplands (P) like Equation (3):

$$P = \{x \in C | x \text{ is top 10th percent}\}. \tag{3}$$

According to Equation (3), the element of potential high-quality croplands is in crop suitability (C). The thresholds were set as the top 10% in the result of crop suitability. Considering the normal distribution, the threshold for the potential high-quality croplands set as 0.94 conservatively. The values exceeding 0.94 are regarded as potential high-quality croplands, whereas values below 0.94 are considered to be unsuitable for potential high-quality croplands.

### 2.3.3. Selecting Climatic and Environmental Variables

With regard to the independent variables, in this study, up to 24 climate variables and 3 environmental variables were listed as shown in Table 1. All climate and environmental variables feature a spatial resolution of approximately 1 km$^2$ (0.00833°). We reviewed previous studies to determine the best-fitting variables in terms of explaining plant growth and croplands, with low statistical inferences and high model reliabilities. The correlation analyses were conducted using the statistical program R studio version (3.6.1). In this study, we excluded climate variables when the correlation coefficient was less than ±0.2 between a dependent variable and 24 independent variables firstly. Secondly, the selected climate variables conducted inter-correlations analysis to examine multicollinearity between climate variables. We selected one variable throughout the literature review if the correlation coefficient is higher than ±0.8. Third, the final selected climate variables were tested throughout the *p*-value to determine statistical significance ($p < 0.005$). The Variance Inflation Factor (VIF) was also utilized for the purpose of validating multicollinearity issues, indicating the multicollinearity problem when VIF is above 5 or 10 [43]. We ensure each variable play an independent role throughout the *p*-value and VIF test.

In terms of the environmental variable, the slope was extracted from the DEM, while the TWI was created from the DEM by considering flow direction and accumulation. ELUs were extracted based on the Kyrgyzstan administrative boundary, using the ArcGIS version (10.6). The data specification of the environmental variables was adjusted to the climate variables, such as resolution (approximately 1 km$^2$), data extension (raster image), and boundaries of the country. Finally, the climatic and environmental data were established in the form of an image file.

2.3.4. Random Forest Model for Assessing Potential High-Quality Croplands

The R packages (R core team, Vienna, Austria, http://www.R-project.org, accessed on 21 July 2021) of "randomForest" were used in this study to identify the potential high-quality croplands in Kyrgyzstan [44]. The point shapefiles of high-quality cropland indicate the labeled data for the random forest model, presenting the location of high-quality cropland with the coordinate of WGS 1984. The model creates decision trees on the randomly-selected 1500 bootstrap samples from the labeled data of high-quality croplands to be predicted by each tree, and selects variables from a list shown in Table 1 [45]. These were combined with the 5 climate and 3 environmental dataset, as needed for the analysis [46]. The absence points were randomly distributed individual points across the target country in this study. The combined points were routinely partitioned into two subsets: Training and validation.

Considering the information from the literature regarding these subsets, we found that 70% of the whole data is common enough for model training, and the rest is often separated to investigate the accuracy of the models' predictions [47]. Meanwhile, the model performance is evaluated by using 30% of testing dataset that was not used for model training. The predictive performance of the models was evaluated by applying a threshold-independent method, the receiver operating characteristic (ROC) curve [48]. The area under the ROC curve (AUC) has been considered to be a quantitative performance metric that explains the accuracy of a model. The model would be considered to exhibit high performance when AUC is closer to 1, but it would be indicated to be a weak performance when the AUC was below 0.5 [49,50]. In terms of analyzing the future potential cropland, we replaced the current climate dataset into future climate datasets, which were generated using the RCP 4.5 and RCP 8.5 scenarios for the 2050s and 2070s, respectively. We assumed that the environmental variables remain unchanged and that identical conditions were be maintained, as climate change was not an influence on these data.

## 3. Results and Discussions

### 3.1. Climate and Environmental Variables for Cropland Suitability Analyses

Among 24 climate variables, 12 uncorrelated variables were excluded resulting from the Pearson correlation coefficient: $r \leq 0.2$ or $r \leq -0.2$. The remaining climate variables were the annual mean temperature (Bio1), isothermality (Bio3), min temperature of coldest month (Bio6), mean temperature of coldest quarter (Bio11), annual precipitation (Bio12), precipitation of wettest month (Bio13), precipitation seasonality (Bio15), precipitation of wettest quarter (Bio16), precipitation of driest quarter (Bio17), precipitation effectiveness index (PEI), aridity index (AI), and climate moisture index (CMI). Later, the best-fitting climate variables were sorted out throughout literature reviews and multicollinearity, wherein the result of Pearson coefficients were greater than 0.8 between the variables (Pearson correlation coefficient; $r \geq 0.8$ or $r \leq -0.8$). As a result, the study categorized 5 climate variables and 3 environmental variables. The selected climate variables are the annual mean temperature (Bio1), isothermality (Bio3), annual precipitation (Bio12), precipitation of the wettest quarter (Bio16), and AI, while the environmental variables included slope, TWI, and ELU. As for environmental variables, we excluded it in the correlation analysis in order to reduce statistical error. The *p*-value and VIF were employed to detect whether the selected variables exhibit multicollinearity with each other. As a result, the selected variables were determined to be statistically significant ($p < 0.05$), and multicollinearity was not observed, given that the value of VIF was less than 10. The best-fitting variables for the running model are delineated, as shown in Table 2.

**Table 2.** Climate and environmental variables.

| Type | Acronym | Full Name | $y \geq \pm 0.2$ | $r \geq \pm 0.8$ | Literature Review | Pr(>|t|) | VIF |
|---|---|---|---|---|---|---|---|
| Climate | Bio1 | Annual mean temperature | 0.22 | $r \geq 0.88$ with Bio6 | [26,51] | $<2e^{-16}$ | 2.032 |
| | Bio3 | Isothermality | 0.2 | $r \geq 0.86$ with Bio6 | [52] | $<2e^{-16}$ | 2.158 |
| Climate | Bio12 | Annual precipitation | 0.32 | $r \geq 0.8$ with Bio11, 15,PEI, CMI | [53] | $<2e^{-16}$ | 1.990 |
| | Bio16 | Precipitation of Wettest Quarter | 0.37 | $r \geq 0.98$ with Bio13 | [54] | $< 5.35e^{-11}$ | 2.881 |
| | AI | Aridity Index | $-0.28$ | $r \geq -0.92$ with Bio17 | [55,56] | $<2e^{-16}$ | 1.393 |
| Environment | Slope | Slope | - | - | [57] | $<2e^{-16}$ | 1.422 |
| | TWI | Topographical Wetness Index | - | - | [32,57] | $<2e^{-16}$ | 1.292 |
| | ELU | Ecological Land Unit | - | - | [58] | $< 1.85e^{-0.5}$ | 1.114 |

### 3.2. High-Quality Croplands in Kyrgyzstan

The results of high-quality croplands were clustered around the Chu-Talas valley, Issyk-kul area, and Fergana valley, considering the upper 5 percent of NDVI (Figure 2a). The high-quality croplands will serve as the labeled data to predict crop suitability. Climatically, the annual mean temperature of suitable cropland in the Chu-Talas valley and Fergana valley are 8 to 16 degree Celsius, while annual average precipitation of three regions is approximately 410 mm to 730 mm. Environmentally, these areas are prone to water accumulation and low slopes that are not over 30°. In addition, areas ecologically located at the cool semi-dry regions that are relatively not too cold or too hot. The high-quality croplands distinctively close to urban areas: Bishkek in the Chu-Talas valley, Osh city in the Fergana valley, and Karakol in the Issyk-kul region.

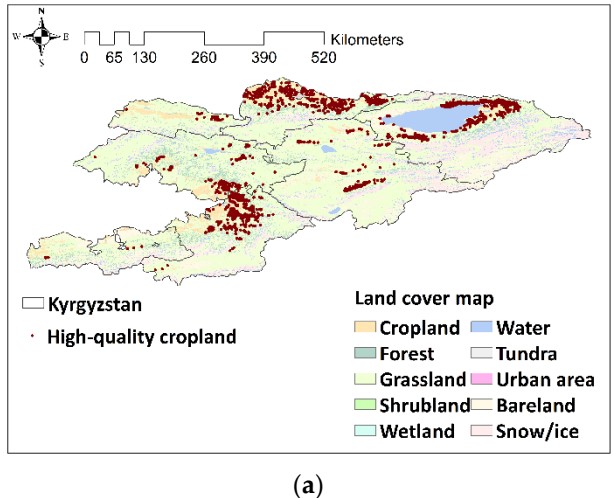

(**a**)

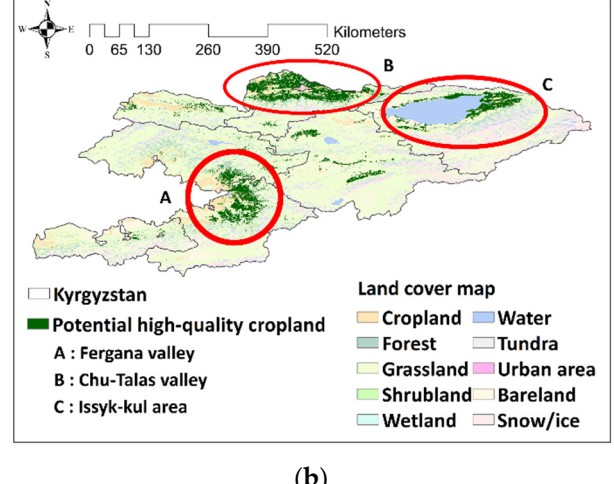

(**b**)

**Figure 2.** High-quality cropland and potential high-quality cropland: (**a**) High-quality cropland in Kyrgyzstan, serving as training label data for predicting potential high-quality cropland. (**b**) Spatial distribution of potential high-quality croplands under baseline climate conditions throughout the random forest model.

### 3.3. Potential High-Quality Croplands under the Baseline Climate

The results indicate that the potential high-quality croplands are clustered around the Chu-Talas valley, Issyk-kul area, and Fergana valley, where surrounding areas of existing croplands are located (Figure 2b). The potential high-quality croplands consisted of croplands (6817 km$^2$, 42.2%), grasslands (7667 km$^2$, 47.4%), forests (943 km$^2$, 5.8%), shrublands (306 km$^2$, 1.9%), wetlands (78 km$^2$, 0.5%), water (37 km$^2$, 0.2%), urban areas (31 km$^2$, 0.2%), and barelands (287 km$^2$, 1.8%). There is no available space for the tundra and snow areas under the current climate conditions (Table 3). The total area of potential high-quality cropland is 16,166 km$^2$, including the high-potential cropland within the cropland category (6817 km$^2$). If these potential high-quality croplands were all converted to the cropland category, the additional area can be extended up to 4.7% (9349 km$^2$) from the forests, grasslands, shrublands, wetlands, water, urban areas, and barelands (Figure 3). The cropland can occupy 12.8% (25,537 km$^2$) of the area of Kyrgyzstan under the land cover map. This area is approximately 1.5 times greater than the original cropland category estimated in 2012.

**Table 3.** Composition of potential high-quality cropland by land use categories.

| Land Category | Land Cover Map (2010) | Potential High-Quality Cropland | |
|---|---|---|---|
| | Area (Km$^2$) | Area (Km$^2$) | Composition (%) |
| Cropland (A) | 16,188 | 6817 (A) | 42.2 |
| Grassland (B) | 112,013 | 7667 (B) | 47.4 |
| Other lands (C) | 71,848 | 1681 (C) | 10.4 |
| -   Forest | 9272 | 943 | 5.8 |
| -   Shrubland | 1032 | 307 | 1.9 |
| -   Wetland | 915 | 78 | 0.5 |
| -   Water | 13,993 | 37 | 0.2 |
| -   Tundra | 51 | 0 | 0 |
| -   Urban | 491 | 31 | 0.2 |
| -   Bareland | 26,268 | 287 | 1.8 |
| -   Snow | 19,825 | 0 | 0 |
| Subtotal (B + C) | 183,861 | 9349 | 57.8 |
| Total (A + B + C) | 200,048 | 16,166 | 100 |

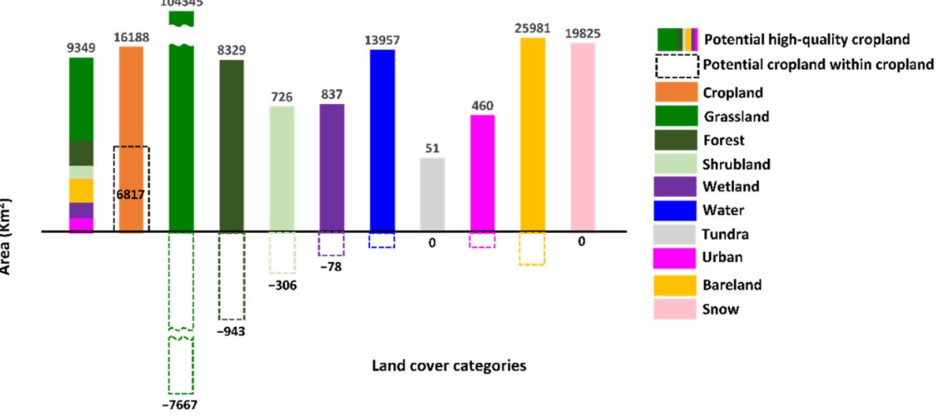

**Figure 3.** Composition of potential high-quality cropland by land use categories.

### 3.4. Validation and Model Performance for Predicting Potential High-Quality Cropland

To examine the accuracy of the model's prediction by the AUC, the validation subset from the high-quality cropland dataset, which was 30% of the dataset, was used. This study exhibited a statistically excellent performance in predicting the most suitable cropland distribution. A total of 95% of the high-quality cropland from the validation set are detected throughout ROC curve, indicating an excellent performance in terms of estimating potential high-quality croplands across the study area. In addition, the results of the RF models indicate the importance of each input variable in relation to the potential high-quality cropland. Annual mean temperature (Bio1) and ELU reveal important variables of the random forest model. A higher score of a variable implies that the variable is more significant. To verify the model performance and original cropland, thresholds were applied (Figure 4). According to conservative thresholds ($\geq$0.94), the predicted potential high-quality cropland was consistent with the cropland designated by the land cover map of Kyrgyzstan (approximately 43.7%), while the predicted potential high-quality cropland was consistent with the original cropland (approximately 30.5%) when this study attempts to set secondary thresholds less conservatively, which refer to the lower 1% of the normal distribution ($\geq$0.74). We classified as mismatched areas when the original cropland does even belong to the lower 1% of the predicted cropland. This mismatch (25.8%) could have been caused by several factors. One possible reason of this mismatch between high-quality cropland and original is the fact that many rural households had small rain-fed plots on steep slopes, which were unequally accessible by land after the land reform process during the collapse of the Soviet Union [59]. This is according to statistical performance and comparison between the predicted cropland and existing cropland. The RF model are reasonable for estimating future suitable cropland [60].

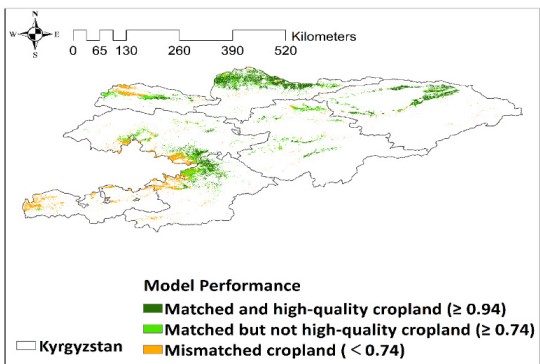

**Figure 4.** Model performance of the random forest model by comparing predicted cropland and existing cropland.

### 3.5. Random Forest Model for Potential High-Quality Cropland under the RCP4.5 and RCP8.5 Scenarios

In the RCP4.5 and 8.5 scenarios, potential high-quality croplands tend to expand inland along the periphery of cropland however, it was difficult to detect high-quality croplands in the 2070s under both scenarios (Figure 5). The primary difference between these two scenarios is the possibility of agricultural activities until 2070s. Generally, the areas show the downward trend. In this study, we identified the area and change in land use proportion of high-quality cropland between the baseline climate and RCP scenarios (Table 4). We categorized three parts as 'cropland remaining potential high-quality cropland (CC)', 'potential high-quality cropland within grassland category (GC)', and 'potential high-quality cropland within other lands category (OC)'. Other lands, here, included forest, shrubland, wetland, water, tundra, urban, bareland, and snow which are all land use proportion except for cropland and grassland. The areas of potential high-quality cropland headed for a sharp decline from 3731.7 km$^2$ in the 2050s to 1793.7 km$^2$ in the 2070s under RCP4.5 scenarios, and the areas of it continually show reduction trends until the 2070s

(914.5 km$^2$) of the RCP8.5 scenarios by climate change (Table 4). The changes in land use proportion of potential high-quality cropland for future climate change in the 2050s and 2070s across RCP4.5 and RCP8.5 scenarios can be compared with the land use proportion of it for the baseline climate in Table 4. For the RCP4.5 scenario, compared to the proportion of potential high-quality cropland under the baseline climate, the change in CC proportion fall −12.1% and −15.9% in the 2050s and 2070s under the RCP4.5 scenarios, while the change in GC proportion is as opposed to the one within CC, showing an increasing trend throughout the RCP4.5 scenario. Furthermore, the proportion of OC has an increase by 1.5% in the 2050s and has a slight decrease by −0.3% in the 2070s. For the RCP8.5 scenario, the CC proportion has a slight increase of 4.0% during the 2050s, but it plunges in the 2070s by −30.4%, while the GC proportion dramatic increases in the 2070s by 30.7%. The proportion of OC drops by half in the 2050s (−5.2%) and continues to decline by −3.0% until 2070. In 2070, of the RCP8.5 scenario, potential high-quality cropland is the largest proportion from the grassland category. Most potential high-quality croplands will be located at the grassland category, which can possibly be changed into croplands in the 2070s.

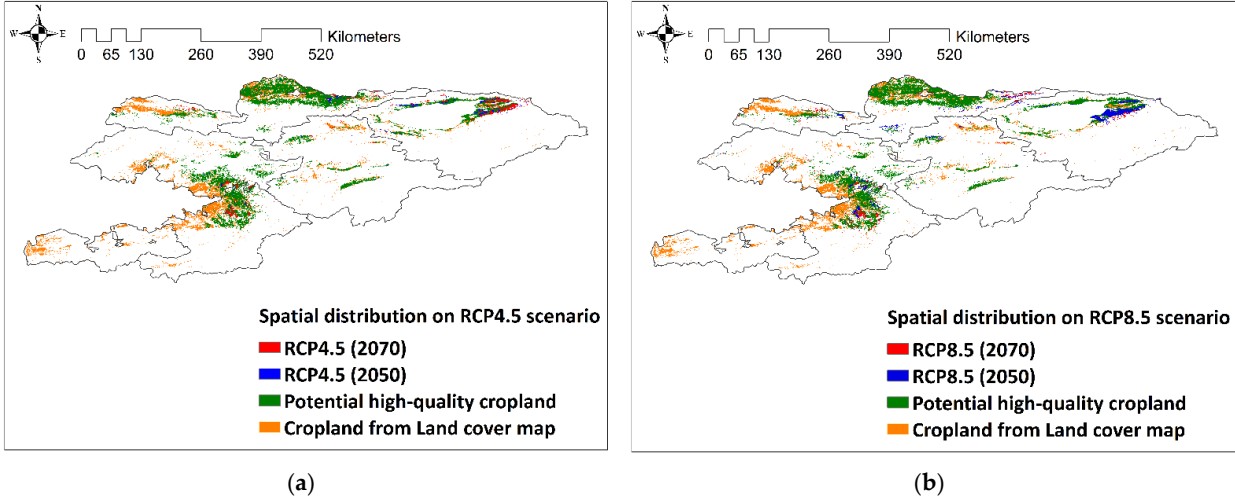

(**a**)          (**b**)

**Figure 5.** Spatial distribution on RCP scenario: (**a**) RCP4.5 scenarios in 2050s and 2070s; (**b**) RCP 8.5 scenario in 2050s and 2070s.

**Table 4.** Land use shares of potential high-quality (H-Q) cropland by different scenarios.

| Scenario | Baseline | | Under the RCP4.5 Scenario | | | | | |
|---|---|---|---|---|---|---|---|---|
| Period | P.H-QCropland (A) | | 2050 (B$_1$) | | | 2070 (B$_2$) | | |
| | A. Area | Prop | B$_1$ Area | Prop | [1] ΔB$_1$ | B$_2$ Area | Prop | ΔB$_2$ |
| Unit | Km$^2$ | % | Km$^2$ | % | | Km$^2$ | % | |
| CC | 6817.5 | 42.2 | 1119.2 | 30.1 | −12.1 | 471.20 | 26.3 | −15.9 |
| GC | 7667.5 | 47.4 | 2153.1 | 58.0 | 10.6 | 1142.1 | 63.7 | 16.3 |
| OC | 1681.0 | 10.4 | 441.3 | 11.9 | 1.5 | 180.4 | 10.1 | −0.3 |
| Total | 16,165.9 | 100 | 3713.7 | 100 | | 1793.7 | 100 | - |
| Scenario | Baseline | | Under the RCP8.5 Scenario | | | | | |
| Period | P.H-QCropland (A) | | 2050 (B$_3$) | | | 2070 (B$_4$) | | |
| | A. Area | Prop | B$_3$ Area | Prop | ΔB$_3$ | B$_4$ Area | B$_4$ | ΔB$_4$ |
| Unit | Km$^2$ | % | Km$^2$ | % | | Km$^2$ | % | |
| CC | 6817.5 | 42.2 | 828.5 | 46.2 | 4.0 | 107.6 | 11.8 | −30.4 |
| GC | 7667.5 | 47.4 | 871.5 | 48.6 | 1.2 | 714.7 | 78.1 | 30.7 |
| OC | 1681.0 | 10.4 | 93.7 | 5.2 | −5.2 | 92.3 | 10.1 | −0.3 |
| | 16,165.9 | 100 | 1793.7 | 100 | - | 914.5 | 100 | - |

P.H-Q cropland—potential high-quality cropland; prop. (%)—proportion; CC—cropland remaining potential high-quality cropland; GC—potential high-quality cropland within grassland category; OC—potential high-quality cropland within other lands category (OC), where other lands is sum of all land use proportion except cropland and grassland's one. [1] $\Delta B_i = B_i - A$ ($\rangle = 1, 2, 3, 4$).

### 3.6. Agricultural Adaptation Strategy for Land-Use Purpose

Kyrgyzstan has been warned regarding the impact of climate change on food insecurity and other environmental issues [61,62]. According to the household food security assessment conducted by the World Food Programme in March 2013, it is estimated that 24% of households suffer from food insecurity. Thus, the government of Kyrgyzstan has adopted national policy and listed ways of improving agricultural infrastructure for land resource management and agricultural adaptation, including integrated soil fertility management, irrigation techniques like drip irrigation, and the use of portable chutes in sloping areas [8]. In terms of agricultural adaptation strategy on enhancing food security, it is important to determine priority areas (Figure 6). The irrigation system can be firstly repaired in low-quality croplands. Although almost 93% of freshwater has been used for agricultural purposes throughout the irrigation channels, more than half of cropland (56.3%) are low-quality croplands. This is attributed to the mismanagement or misallocation of croplands with climatic factors [63]. In addition, in determining additional croplands, there are other options for the adaptation strategy by considering the type of land use. Grassland to cropland can be one approach since Kyrgyzstan has experienced considerable reductions in cropped areas and corresponding increases in grasslands, which result from the discontinuation of rain-fed crop production by land reform after the dissolution of the Soviet Union [8]. Another approach is to extend croplands by reverting the barelands to cropland. It is necessary to assign the land use purpose and manage the barelands accordingly, because these can undergo further degradation when abandoned [64].

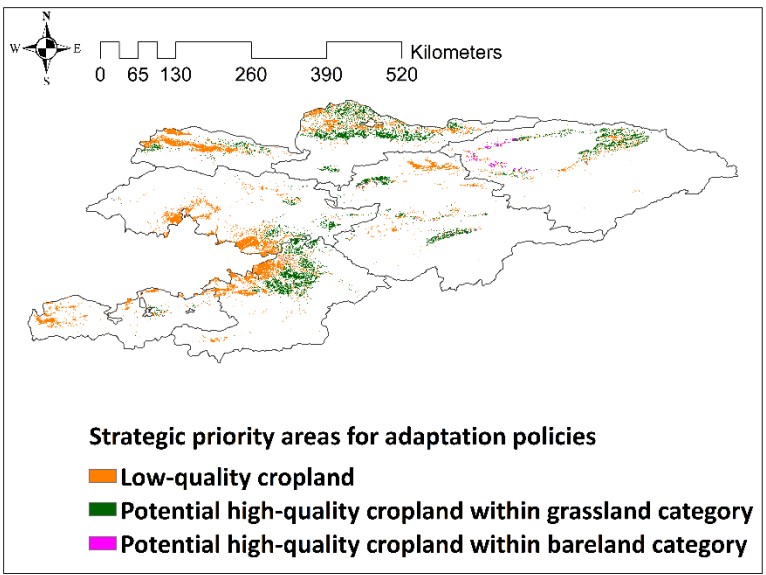

**Figure 6.** Strategic priority regions for agricultural adaptation policies.

### 3.7. Agricultural Adaptation Strategy against Climate Change

The threats of climate change far outweigh the opportunities in agriculture, considering the short- and long-term impacts [65]. Cooler regions appear to benefit, i.e., an increase in the temperature of arable areas [66], whereas the receding glaciers pose threats for irrigation-dependent agriculture in both short- and long-term considerations [55]. In Kyrgyzstan, such as those in cooler regions, climate change may temporarily pose opportunities due to the suitable temperatures. However, croplands are expected to be irreclaimably decreased in the far future. The size of the cropland dramatically decreases in the 2070s under the RCP8.5 scenario, and the increased temperatures pose a deadly threat to general crop growth during flowering [57]. The relationship between temperature change and crop yield can be observed in other Asian countries, where it is argued that an increase of 1 °C during wheat growing season reduces the wheat yields by approximately 3–10% [67]. This scientific evidence reinforces the future scenarios. Thus, it is inevitable to

establish the agricultural adaptation strategy against climate change based on the climatic and environmental conditions.

In this sense, it is necessary to identify the target croplands for the appropriate implementation of agricultural resilience policies. In terms of efficiency and sustainability, an agricultural adaptation strategy should be adopted in certain regions that possess climatic–environmental potential high-quality croplands and in areas with cropland that remain as potentially high-quality croplands even under the RCP4.5 and 8.5 scenarios. There are few regions that satisfy these abovementioned conditions. These priority regions pave the pathway for the future as potential areas where policies for agricultural resilience can be enabled and established according to different periods and scenarios. The best-fit regions, where high-quality croplands remain as potential high-quality croplands until the 2050s and 2070s, are shown in Figure 7.

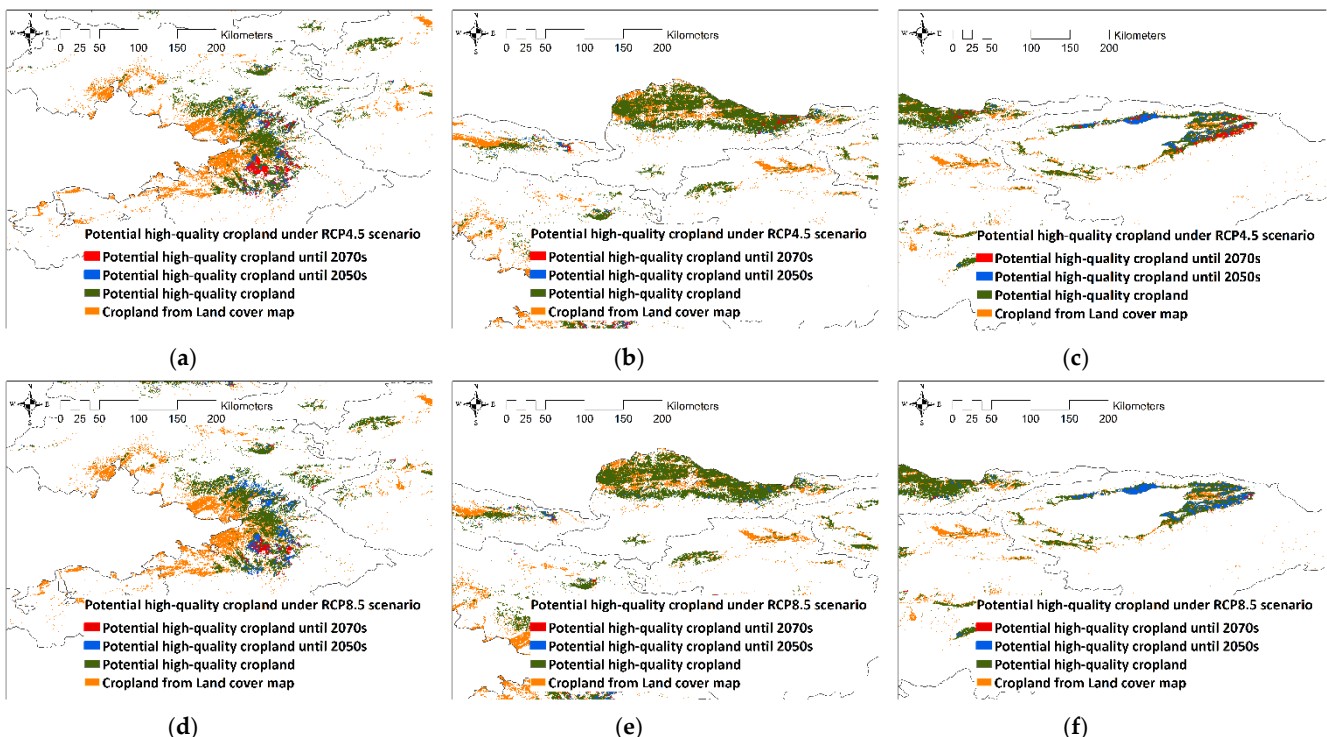

**Figure 7.** Spatial distribution of future potential high-quality croplands under different scenarios: (**a**) Fergana valley in RCP4.5; (**b**) Chu-Talas valley in RCP4.5; (**c**) Issyk-kul area in RCP4.5; (**d**) Fergana valley in RCP8.5; (**e**) Chu-Talas valley in RCP8.5; and (**f**) Issyk-kul area in RCP8.5.

### 3.8. Implications of Proactive Policies on Food Security and Agricultural Resilience

The proactive policies on food security and agricultural resilience have implications on achieving global goals. Kyrgyzstan has achieved Goal 1 of the Millennium Development Goals (MDGs), which is to halve the extreme poverty level and to halve the proportion of people who suffer from malnourishment [68]. Although the country has reached and exceeded its target extreme poverty level, the level of poverty remains significant, even though the poverty line used in Kyrgyzstan is considerably low. Kyrgyzstan has made the transition from the MDGs to the SDGs. However, the country still faces significant challenges in its pursuit of zero hunger [69]. Although the zero hunger indicator in Goal 2 has steadily declined since 2000, 6.5% of the population has still been suffering from hunger in 2016, and 12.9% of children under 5 years of age had stunted growth in 2014 (SDG country profile assessed on 14 May 2020). According to the government resolution, 48% of the population living in Jalal-Abad oblast were struggling with securing sufficient food, followed by people living in Osh Oblast (38%) and Batken Oblast (38%) [6]. Therefore, the proactive approach on food security and agricultural resilience needs to be coherent

with policies in relation to global goals by reinforcing equality vis-à-vis climate changes. Moreover, the short- and long-term policies should highlight the implementation of SDGs by considering vulnerable groups, given that the goal of zero hunger is clearly aimed at the poor as well as the people subjected to vulnerable situations [70].

## 4. Conclusions

Kyrgyzstan is a vulnerable country to climate change impacts, especially in the agricultural sector. This study aims to determine the potential high-quality cropland and future potential high-quality cropland using the random forest model, to establish agricultural adaptation policies. In terms of training data, maximum cropland's NDVI were used for representing high-quality croplands, while five climate variables and three environmental variables throughout the correlation coefficient and multicollinearity were selected during the regression analyses. In analyzing the potential high-quality cropland, present climate data were analyzed; in terms of future high-quality cropland, different future climate data were utilized: Near future (2050s) and far future (2070s) in the RCP4.5 and RCP8.5 scenarios, respectively. This was done under the assumption that there will be no changes in environmental variables. The study determined that potential high-quality croplands (9349 km$^2$) from the forests, grasslands, shrublands, wetlands, water, urban areas, and barelands. There potential croplands are clustered around three main regions: The Chu-Talas valley, Fergana valley, and Issyk-kul area. The cropland category can be extended by 1.5 times compared to 2012, if all potential high-quality croplands converted to the cropland category. However, these potential high-quality croplands are expected to expand toward the inland in the future, by potentially changing land use from grasslands to croplands. The proportion of potential high-quality cropland was determined to gradually decrease due to climate change. However, the portion of high-quality croplands is expected to increase in grasslands, indicating that original croplands may not need as much agricultural activities compared to the past. Agricultural adaptation strategies are necessary, considering potential high-quality croplands. Agricultural infrastructure should be improved with targeting low-quality croplands and high-quality areas within grasslands. Taking action against climate change, agricultural resilience can be applied across the target areas where potential high-quality croplands as well as future high-quality cropland are both satisfied. These proactive agricultural policies on food security and overall resilience must be geared toward achieving zero hunger among vulnerable groups.

**Author Contributions:** Writing, S.P.; methodology, C.-H.L. and S.J.K.; validation, S.-E.C. and E.I.; formal analysis, S.-D.L.; supervision, W.-K.L. All authors have read and agreed to the published version of the manuscript.

**Funding:** This research was funded by the Korea Agency for Infrastructure Technology Advancement (KAIA), grant funded by the Ministry of Land, Infrastructure, and Transport, grant number 20UMRG-B158194-01.

**Institutional Review Board Statement:** Not applicable.

**Informed Consent Statement:** Not applicable.

**Data Availability Statement:** Not applicable.

**Conflicts of Interest:** The authors declare no conflict of interest.

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
