# Peer review of "Assessing Climate Change Impact on Cropland Suitability in Kyrgyzstan: Where Are Potential High-Quality Cropland and the Way to the Future"

_agronomy, doi:10.3390/agronomy11081490_

Round 1

Reviewer 1 Report

Dear authors,

The article deals with very important topic today, the relationship between climate change and agriculture.

There are several points of the article that should be improved:

  • In the row 77 (introduction) mention which wheat (autumn or spring or both). 
  • In point 2.1, I suggest you to make a map of the study area, where to find the altitude steps, the four climatic zones and the Köppen classification. Also, on this map, in the medallion to be a map of the geographical location of the study area. A map helps the reader better follow the written information.
  • List the climate variables (for example: mean, maximum and minimum temperatures etc.) and the five indices in sections 2.2.2. and. 2.3.3.
  • Better edit Figure 1, with the same format for editing and an easier-to-read legend (the writing is not legible).
  • The maps from figure 1 do not identify the three areas mentioned in the text (Chu-Talas valley etc.). Fix this issue on maps.
  • Between values of precipitation and mm a free space is left. 
  • I suggest you make a table for the values mentioned for the potential high-quality croplands composite in section 3.3. They are difficult to follow in text format.
  • The writing in figure 2 is not legible.
  • Figure 3 has several aspects to improve. Highlight in the figure the study area by a delimitation in a red border for example. Legibly written in the legend. 
  • Withdraw the first row to the paragraph in section 3.5.
  • The writing in figure 4 is not legible.
  • Figure 5 has several aspects to improve. Highlight in the figure the study area by a delimitation in a red border for example. Legibly written in the legend.
  • There are the same issues that need to be improved in Figure 6.
  • You need to renumber the sections. You do not have section 4 in the article.

I wish you all the best!

Author Response

We appreciate your positive and careful review. We have endeavored to improve the value of the manuscript by revising the specific part you pointed out. We revised all the figures to make it clear, also we revised throughout the paper, including data and methodology parts. Please kindly check attached word file

Reviewer 2 Report

I'm reviewing "Assessing Climate Change Impact on Cropland Suitability in Kyrgyzstan: Where are the Potential High-Quality Cropland and the Way to the Future".
The topic considered in the paper is of interest for the reader.
Some parts could benefit a language revision.

There is need of revisions at lines:
186 A citation is needed in order to use NDVI for classifying croplands

217 Does the independent variables selection occur before the RF modelling? RF is supposed robust against multicollinearity so that it should run on the whole set.

142-144 using the max(NDVI_year) as proxy of crops yield should be supported by a description of main local crop system (how many crops per year, rotation scheme,...).

191-196 NDVI ranges from -1 to 1. Do you adopt NIR/VIS simple ratio?

228 R-squared ranges between 0 and 1

246-248 if climate and environmental datasets are completed with the RF output, what are the variables the RF is running on?

I would like to suggest some improvement at lines:
65 "agricultural infrastructure" would be clearer if an example were provided. I.e., does the action focus on road for agricultural machinery or sewer for watering?
70 as previous, which "technology" are under consideration? More fitting seeds genetics? Drip irrigation? VRT fertilization?

139 nearest neighbor should classify the 250m pixel the same as the "center" (in the ~8x8 pixel_30m within the superposed pixel_250m would be considered the 4,4), isn't it?
146-150 While upsampling a matrix the pureness of "cropland" within pixel_250m could be computed, as well as the relative abundance of other classes in order to clear the dataset.  As it is exposed here, I do not understand the background. Did you remove the "part forest part cropland" pixel due to known mixture in Kyrgyzstan's landscape? What about mixtures with other classes?

145 does "where have information" mean the same as "where information is available"?

155-163 I expected a table for the climate data and indices considered, or at least a more in deep mention than "based mainly on the monthly average temperature and precipitation". The same way the environmental indices are proposed in 2.2.3 would fit. You expose in 3.1 the 12 variables used; this should be mentioned here.

139-141 Not sure of what I'm reading. Do you explain that you spatially subset the NDVI products on the cropland mask? Did you adopt some NDVI pattern as "cropland" classification?

191-196 A further consideration about the methodological approach adopted (in link with the crop-system details). NDVI timeseries based on quantiles of the scored max(NDVI) could be inspected. I.e. NDVI timeseries can be plotted NDVI-time exposing the classification bounds for "high-quality croplands" (maybe a quantile based sampling can enhance the readability of the graphic output).

324-326 The stated result is very evocative but could be formulated in a straighter manner. "95% of real high-quality cropland from the validation set are detected".

350 Readers would gain in readability if the sites you mention could were labeled in corresponding figures.

419 from Table2 it seems that specifically targeted conversions from grassland to cropland could preserve the amount of high-quality cropland area (ΔBx absolute values). Is this wrong?

Author Response

We appreciate your positive and careful review. We have endeavored to improve the value of the manuscript by revising the specific part you pointed out. We revised all the figures to make it clear, also we revised throughout the paper, including data and methodology parts.

Round 2

Reviewer 1 Report

Dear authors,
I appreciate your effort to improve the article.

However, there are still small issues to complete as well:

  • Use the same name for the climate classification in the legend of figure 1, as in the text: the Köppen classes. The same change for the title of figure 1: Köppen climate classification. Or use the name everywhere (text, legend and title of figure 1): Köppen-Geiger climate classification. You decide which name of climate classification to choose!
  • In table 1, I propose for Bio15 climate variable to explain what C of V means. It is understood that it is a unit of measurement ... but, which one? Add a note to the table to explain the acronyms. Put the acronyms of the environmental variables, after the full names.
  • You have no reference in the text for figure 3. You have to add it.
  • Figure 7 is actually figure 6. Check the numbering of the figures.

I wish you success in your endeavor to publish the article.

Sincerely your!

Author Response

We very appreciate your endless support and effort to the review process. We revised the small issues in the articles as your comments. Please kindly check attached file. 

Reviewer 2 Report

I'm reviewing the 2nd version of the manuscript "Assessing Climate Change Impact on Cropland Suitability in Kyrgyzstan: Where are the Potential High-Quality Cropland and the Way to the Future".
The manuscript improved sine the first submission. The current version includes fixes and is more detailed in both methods and results presentation.
The proposed manuscript is suitable for publication on MDPI Agronomy.

Author Response

We would like to express our gratitude to you who kindly reviewed and provided insightful comments & suggestion for improving our manuscript. Thanks to your comment, we believe this manuscript can have an opportunity to publish on MDPI Agronomy.